# Beyond Perplexity: UTF-8 Validity in Byte-aware Language Models

**Sangwhan Moon**[1]  **Daisuke Oba**[2]  **Youmi Ma**[2]  **Tatsuya Hiraoka**[3]  **Naoaki Okazaki**[2]

## Abstract

Byte-level tokenization enables language models to handle any Unicode input, but models can generate invalid UTF-8 sequences when encountering rare or unseen characters. We investigate the relationship between training scale and UTF-8 generation reliability with a 355M parameter model trained on 80B tokens from a balanced multilingual corpus of English, Japanese, Korean, and Chinese. We introduce multiple evaluation protocols that isolate UTF-8 structural validity from language modeling. UTF-8 validity convergence lags perplexity by roughly a factor of two: perplexity stabilizes after 2.1B tokens, but UTF-8 validity requires 4.2B tokens. In context-free generation, common characters achieve higher structural validity than rare characters, with byte-length exposure emerging as an additional axis of difficulty alongside frequency. Our experiments show that reliable UTF-8 generation is a distinct capability requiring evaluation beyond perplexity.

 github.com/cynthia/bytecanary

## 1. Introduction

Multilingual NLP systems inevitably face unknown characters: limited vocabulary budgets prevent full Unicode coverage, causing tokenizers to fail on characters outside their vocabulary. This problem often occurs when handling languages that use non-Latin alphabets, such as CJK languages. To mitigate this, most popular large language models (LLMs) utilize byte-fallbacks that includes tokens corresponding to all bytes so that the tokenizer can encode texts with unknown characters.

Beyond the byte-fallback, some researchers have attempted to develop models with byte-level tokenization, which allows tokens to be separated by byte boundaries rather than character boundaries (Gillick et al., 2016). Notable examples include ByT5 (Xue et al., 2022), which demonstrated competitive performance operating directly on UTF-8 bytes, and the Byte Latent Transformer (BLT) (Pagnoni et al., 2025), which achieved comparable results to Llama 3 while using fewer inference FLOPs. The advantage of byte-level tokenization is that we can flexibly tokenize multi-byte characters into meaningful minimal units. For example, we can extract a primary radical from a single hanzi/kanji character or a single jamo character into pronunciation units. This flexibility enables accurate multi-byte character understanding, effective token embedding learning, and improved downstream task performance (Xue et al., 2022).

Despite these advantages, byte-level tokenization also has a problem in its decoding phase. Since the vocabulary in byte-level tokenization allows tokens that begin or end with a byte in the middle of characters, an NLP system sometimes generates an invalid Unicode sequence, where some tokens cannot connect with each other appropriately. This damages performance in NLP tasks that require models to generate texts, such as machine translation (Wang et al., 2020). Prior work in the LLM field has not thoroughly discussed this problem, likely because byte-level tokenization has been deployed primarily in sufficiently large systems capable of learning valid Unicode sequences. However, considering the difficulty of learning valid Unicode sequences, we assume that byte-level tokenization requires significantly more training data or trainable parameters so that the model can sufficiently learn to generate valid sequences.

We investigate this by training a 355M parameter GPT-2 architecture on 80B tokens of multilingual data comprising English (10%) and Japanese, Korean, and Chinese (30% each), evaluating UTF-8 validity across 420 training checkpoints. UTF-8 validity convergence lags perplexity convergence by a factor of two: perplexity stabilizes after 2.1B tokens, but UTF-8 validity requires 4.2B tokens. Common characters achieve higher structural validity than rare ones (96.21% vs. 95.26%); a control experiment further shows that byte-length exposure is a major axis of difficulty alongside character frequency. Structural validity also exceeds semantic correctness—Term Match Rate reaches 60.30% despite high validity—suggesting that generating a *valid* character is easier than generating the *correct* one. These results

[1]Google LLC / Mountain View, CA, USA [2]Institute of Science Tokyo / Tokyo, Japan [3]Mohamed bin Zayed University of Artificial Intelligence / Abu Dhabi, United Arab Emirates. Correspondence to: Sangwhan Moon <sangwhan@iki.fi>.

*Proceedings of the 43rd International Conference on Machine Learning*, Seoul, South Korea. PMLR 306, 2026. Copyright 2026 by the author(s).

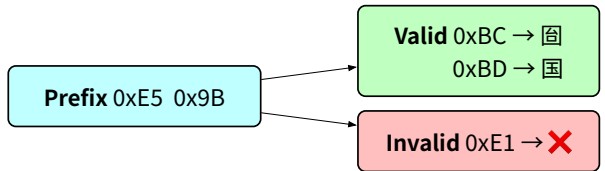

*Figure 1.* An example of valid vs. invalid byte sequences. The invalid case cannot be decoded by a UTF-8 codec.

have practical implications: models that appear well-trained by perplexity may still produce invalid UTF-8 sequences in context-sparse generation.

## 2. Problem Statement

Modern large language models face a fundamental challenge in generating valid Unicode byte sequences, particularly when encountering rare or unseen characters. While byte-level tokenization offers theoretical advantages over character-based approaches—including the ability to flexibly tokenize multi-byte characters and handle any possible input—it introduces a critical failure mode: models can generate invalid UTF-8 sequences that violate Unicode encoding constraints (Figure 1).

To understand this problem, consider how UTF-8 encoding works. Each Unicode character is encoded as a sequence of one to four bytes following strict patterns. ASCII characters (U+0000 to U+007F) use a single byte with the pattern `0xxxxxxx`. Characters from U+0080 to U+07FF require two bytes following `110xxxxx 10xxxxxx`, while characters from U+0800 to U+FFFF need three bytes as `1110xxxx 10xxxxxx 10xxxxxx`. Finally, characters from U+10000 to U+10FFFF are encoded with four bytes following `11110xxx 10xxxxxx 10xxxxxx 10xxxxxx`. Each continuation byte must begin with `10`, and the leading byte determines how many continuation bytes follow. When a model trained with byte-level tokenization generates text, it must implicitly learn these encoding rules to produce valid sequences. However, this learning depends critically on exposure to diverse byte patterns during training.

The problem becomes particularly acute in the context of the long-tail character distribution prevalent in natural language. Consider a rare character like U+2B740 (CJK ideograph) encoded as `0xF0 0xAB 0x9D 0x80`. If this character appears with probability $p(c) < \frac{1}{N}$ (where $N$ is the total number of tokens in training data), the model may never observe this specific byte sequence. When prompted to generate text following a prefix containing `0xF0 0xAB`, the model must correctly predict that the next byte must match the pattern `10xxxxxx` as a continuation byte, i.e., `0x9D` would be valid while `0xF0` would not, and that after `0x9D`, another continuation byte is required. Without sufficient exposure to similar patterns, the model might generate `0xF0 0xAB`

`0xF0`—starting a new 4-byte sequence instead of continuing the current one. This creates an invalid UTF-8 sequence that cannot be decoded, causing downstream applications to fail with decoding errors or produce replacement characters.

The severity of this problem extends beyond simple decoding failures. When models generate invalid sequences, they can enter unstable states where subsequent generation becomes increasingly incoherent. We hypothesize that these failure modes can be triggered adversarially by crafting inputs with rare byte sequences, potentially causing models to generate streams of invalid bytes that appear as corrupted text, fall into repetitive patterns trying to "escape" invalid states, produce outputs that bypass safety filters due to tokenization confusion, or exhibit degraded performance on downstream tasks when rare characters appear.

The core research questions we address are:

- What is the empirical relationship between training scale and UTF-8 generation reliability?
- How does UTF-8 validity convergence relate to perplexity convergence during training?
- Do rare and common characters exhibit different validity learning patterns?
- Does semantic context affect UTF-8 validity learning?

## 3. Evaluation Framework

Byte-fallback tokenization allows language models to represent arbitrary Unicode strings, but generation can still fail by emitting byte sequences that are not valid UTF-8. Crucially, such failures are only weakly coupled to standard language modeling metrics: a model can assign high probability mass to structurally invalid continuations, and perplexity can stabilize while UTF-8 validity continues to improve. We thus propose to evaluate *UTF-8 generation reliability* as a distinct capability along three dimensions.

Given a prompt $C$ and a generated token sequence $x_{1:T}$, let $\text{Bytes}(x_{1:T})$ denote the corresponding byte stream after detokenizing (including bytes produced by byte-fallback tokens). We measure:

**(G1) Structural validity.** Whether $\text{Bytes}(x_{1:T})$ forms a valid UTF-8 string (and at which step it fails if not).

**(G2) Semantic correctness (when a target is defined).** Whether the model outputs the *correct* byte sequence for a particular OOV character, not merely *some* valid UTF-8.

**(G3) Probabilistic preference.** When greedy sampling fails to output the correct bytes, determine whether the model assigns higher likelihood to the correct completion.

We evaluate these axes with two complementary settings, Level 0 and Level 1, and a metric suite covering structural, semantic, and probabilistic behavior.

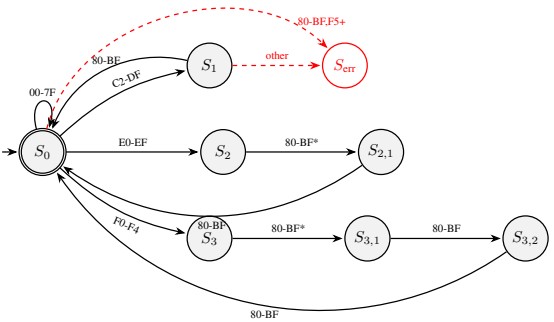

*Figure 2.* Simplified UTF-8 DFA with dashed red error transitions.

*Table 1.* Metric roles in our framework. † Indirect: perplexity reflects average predictive fit, not pairwise preference between gold and generated completions.

| Metric | Structural G1 (§ 3.2.1) | Semantic G2 (§ 3.2.2) | Diagnostic G3 (§ 3.2.3) |
|---|---|---|---|
| $V_{\text{partial}}$ (Eq. 1) | ✓ | | |
| Term Match (Eq. 4) | | ✓ | |
| $\Delta_{LL}$ (Eq. 5) | | | ✓ |
| Perplexity / NLL | | | ✓† |

## 3.1. Evaluation Tasks

**Level 0: Context-Free Structural Validity.** Level 0 isolates UTF-8 structural reliability from language understanding. We prompt the model with prefixes that contain rare or unseen characters (under byte-fallback), generate continuations, and score the produced byte streams by UTF-8 validity. To probe generalization, we stratify targets by character frequency tiers (e.g., common or uncommon).

**Level 1: Context-Guided Byte Retrieval.** Level 1 tests whether semantic and syntactic context can guide the model to retrieve and complete an OOV character's byte sequence. Given a sentence $S$ containing a target character $c$, we form a prefix that includes the context preceding $c$ and a partial byte prefix of $c$. The model must generate the remaining bytes of $c$ as the immediate continuation. This task explicitly couples semantics to byte-level output while keeping the evaluation local to a short completion window.

## 3.2. Metrics

Our evaluation targets three distinct objectives: (G1) *structural* UTF-8 validity of the generated byte stream, (G2) *semantic* correctness when a gold target character is defined (Level 1), and (G3) *diagnosis* of whether errors arise from missing knowledge or from decoding/calibration. No single metric captures all three of them, so we report a complementary suite (Table 1).

### 3.2.1. STRUCTURAL VALIDITY VIA UTF-8 DFA

UTF-8 validity is a property of the *detokenized byte stream*. Let $x_{1:T}$ be the generated token sequence and $B = \text{Bytes}(x_{1:T})$ the corresponding bytes (including byte-fallback tokens). Since UTF-8 defines a regular language over bytes, structural validity can be checked exactly by a deterministic finite automaton (DFA; Figure 2): state $S_0$ represents a character boundary, intermediate states represent partially emitted multi-byte characters awaiting continuation bytes, and $S_{\text{err}}$ denotes an invalid transition. We enforce standard UTF-8 constraints, rejecting overlong encodings, surrogate halves, and codepoints above U+10FFFF; see Appendix E for full transition details.

**Partial credit validity.** Binary validity is brittle when generation stops mid-character (structurally consistent but incomplete). To separate *incomplete* from *invalid* outputs, we compute a DFA-based partial-credit score. Let $b_c$ be the number of bytes belonging to complete valid characters; let $b_i$ be the number of bytes in the trailing (possibly incomplete) character; and let $p \in [0, 1]$ denote the fractional progress within that trailing character according to the DFA. We define:

$$V_{\text{partial}}(B) = \frac{b_c + p \cdot b_i}{|B|}. \tag{1}$$

**Aggregation across generation steps.** Per-step structural scores fluctuate during multi-byte emission (e.g., $0.33 \to 0.67 \to 1.0$ under $V_{\text{partial}}$ for a 3-byte character). For stable monitoring across checkpoints and lengths, we aggregate prefix-wise scores computed on $B_t = \text{Bytes}(x_{1:t})$. We report the running mean:

$$V_{\text{cumulative}}(t) = \frac{1}{t} \sum_{i=1}^{t} V_{\text{partial}}(B_i), \tag{2}$$

and, when emphasizing recent behavior, an exponential moving average:

$$V_{\text{ema}}(t) = \alpha V_{\text{partial}}(B_t) + (1 - \alpha) V_{\text{ema}}(t - 1). \tag{3}$$

### 3.2.2. SEMANTIC CORRECTNESS VIA TERM MATCH

UTF-8 validity (§ 3.2.1) does not imply the model produced the *intended* character. In Level 1, we define a gold byte completion for a target character. Let the target bytes be $B_c = (B_p, B_r)$, where $B_p$ is the provided byte prefix and $B_r$ is the remaining suffix to be generated. We report a binary Term Match indicator that is satisfied only if the model emits exactly $B_r$ as the immediate continuation *and* returns to a UTF-8 boundary after consuming it:

$$M = \mathbb{I}\big[\text{the next bytes complete } B_r$$
$$\text{and the DFA state returns to } S_0\big]. \tag{4}$$

This excludes cases that are UTF-8 valid but correspond to a different character.

### 3.2.3. DIAGNOSIS VIA LIKELIHOOD COMPARISON

To distinguish missing knowledge from decoding/calibration failures, we compare teacher-forced log-likelihoods of the gold completion versus the generated completion under the same context $C$. For gold token sequence $X_{\text{gold}}$ and generated sequence $X_{\text{gen}}$, we compute:

$$\Delta_{LL} = \sum_{t=1}^{|X_{gold}|} \log P_\theta(x_t^{gold} \mid C, x_{<t}^{gold})$$
$$- \sum_{t=1}^{|X_{gen}|} \log P_\theta(x_t^{gen} \mid C, x_{<t}^{gen}) \qquad (5)$$

$\Delta_{LL} > 0$ indicates the model prefers the correct completion even when the decoder fails to emit it (suggesting decoding/calibration issues); $\Delta_{LL} < 0$ indicates the model confidently prefers an incorrect continuation.

### 3.3. Evaluation Protocols

We evaluate each saved checkpoint on fixed *trial sets* of $M = 256$ samples per language for both Level 0 and Level 1, enabling learning curves with minimal evaluation variance. Unless otherwise noted, we use the same decoding configuration across checkpoints and compute structural, semantic, and preference metrics as defined in Sec. 3.2. Full construction details are provided in Appendix I and J.

**Level 0: frequency-tiered OOV characters.** We construct a trial set $D_{\text{trial}}$ of OOV characters stratified into four frequency tiers (*Common/Uncommon/Rare/Unseen*) to control difficulty. We define the set of *seen* characters as $K = V \cup S$, where $V$ is the set of Unicode characters covered by tokenizer vocabulary tokens and $S$ is the set of OOV characters observed in the training corpus under byte-fallback. The *Unseen* tier is sampled from $U \setminus K$ for a predefined Unicode universe $U$ (details in Appendix I). To enable direct comparability between context-free and context-guided settings, the *Common* tier is chosen to overlap with the Level 1 target pool. We use script-aware stratification to avoid mono-script tiers (Appendix I).

**Level 1: context-guided byte completion.** We extract OOV target characters by scanning the pre-tokenized training stream for contiguous byte-fallback sequences and decoding them to UTF-8. Level 1 evaluation focuses on *Common* tier characters; synthetic context generation for *Rare* and *Unseen* characters proved unreliable because target usage in natural-sounding sentences was difficult to validate automatically. To construct controlled contexts without reusing pre-training text, we generate single-sentence prompts using Gemini 3 Pro and filter them for language correctness and uniqueness (Appendix J). Given a sentence containing target character $c$, we apply the *Sentence Prompt*

constraint by providing the preceding context $C_{\text{ctx}}$ and a short byte prefix $B_p$ of the target bytes $B_c$; the model is evaluated on whether it emits the remaining suffix $B_r$ as the immediate continuation (Sec. 3.2.2).

## 4. Experimental Setup

### 4.1. Model and Tokenizer

We train a 355M-parameter decoder-only Transformer based on a GPT-2-style architecture. Implementation details of the model are explained in Appendix B. We use an 8,000-token BPE vocabulary with byte-fallback for out-of-vocabulary characters. When a character is not covered by the vocabulary, it is encoded as its UTF-8 byte sequence using dedicated byte tokens (e.g., <0xE4><0xB8><0xAD> for 中). This preserves information for arbitrary Unicode input at the cost of sequence length.

### 4.2. Training Data

We construct a multilingual corpus from FineWeb (Penedo et al., 2024) for English and FineWeb2 subsets for Japanese, Korean, and Simplified Chinese. The target token ratio is 10% English and 30% each for Japanese/Korean/Chinese. To match this ratio without truncating text, we employ Weighted Dynamic sampling, which preserves natural document boundaries while converging to the target distribution.

The method uses an adaptive weight adjustment with exponential correction based on distribution deviation:

$$w_l^{t+1} = w_l^t \cdot f(\delta_l) \qquad (6)$$

$$f(\delta_l) = \begin{cases} 1 + \alpha \cdot (e^{|\delta_l| \cdot \beta} - 1) & \text{if } \delta_l < 0 \\ \frac{1}{1+\alpha \cdot (e^{|\delta_l| \cdot \beta} - 1)} & \text{if } \delta_l > 0 \end{cases} \qquad (7)$$

$$\delta_l = \frac{\text{actual}_l - \text{target}_l}{\text{target}_l} \qquad (8)$$

where $w_l^t$ is the weight for language $l$, $\delta_l$ is the relative deviation, and $\alpha, \beta$ are hyperparameters controlling adjustment aggressiveness. Alternative sampling methods were evaluated and rejected; see Appendix C for details.

### 4.3. Optimization and Compute

We train with AdamW ($\beta_1 = 0.9$, $\beta_2 = 0.95$, weight decay 0.1) and a cosine learning-rate schedule with peak learning rate $3 \times 10^{-4}$ and 2,000 warmup steps. The global batch processes 5.64M tokens per step across 8 GPUs. Training runs for 14,189 steps (80B tokens; one epoch over the sampled stream), and we save checkpoints every 20 steps (112.8M tokens). For tractable checkpoint-wise evaluation we subsample the saved checkpoints, yielding the 420 evaluated checkpoints reported in the results. Compute budget details are in Appendix B.

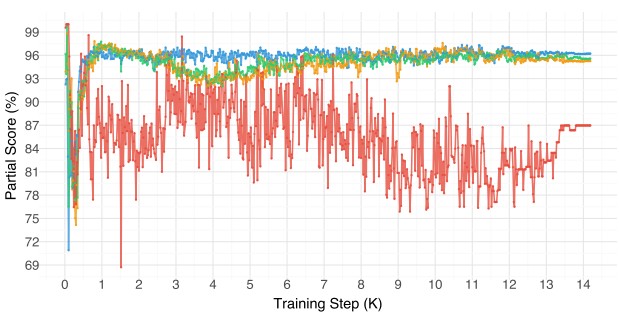

*Figure 3.* Per-tier plots for partial-credit validity. Common (blue), Uncommon (green), Rare (orange), Unseen (red).

# 5. Results

Two evaluation protocols isolate UTF-8 generation capability from general language modeling. Level 0 tests completion of valid UTF-8 byte sequences without contextual cues; Level 1 tests retrieval of correct byte sequences when semantic context constrains the target character. Results span 420 training checkpoints from step 20 to step 14,189.

## 5.1. Level 0: Context-Free Evaluation

Level 0 evaluation tests the model's ability to generate valid UTF-8 continuations given only a partial byte sequence prefix, without any linguistic context to guide its predictions. While this setting is relatively rare in naturalistic text generation—rare characters seldom appear at the absolute beginning of a sequence without any preceding context—it isolates whether the model has internalized UTF-8 structural constraints independent of semantic knowledge.

We evaluated the model using 2-byte prefixes, which we found to be more diagnostic than single-byte prefixes. Single-byte prefixes provide insufficient constraint, as the model can satisfy UTF-8 validity requirements through multiple valid continuation patterns. With 2-byte prefixes, the model must correctly identify whether the sequence requires additional continuation bytes and, if so, generate bytes matching the required 10xxxxxx pattern.

### 5.1.1. FREQUENCY AND VALIDITY

Character frequency and structural validity exhibit a clear relationship in Figure 3. At the final checkpoint (step 14,189, corresponding to 80B tokens), the model achieved its highest partial-credit validity rates on the *Common* tier (96.21%), followed by the *Uncommon* tier (95.57%), with the *Rare* tier close behind at 95.26%. The *Unseen* tier, containing characters never observed during training, achieved 86.97% partial-credit validity. We use $V_{partial}$ as a learning-dynamics diagnostic throughout the main results; strict validity, which most deployed applications require, remains substantially lower (cf. Table 3).

*Table 2.* Partial-credit validity (%) on unseen 4-byte vs. 3-byte characters across prefix lengths; prefix=3 crossover is structural.

|  | Prefix=1 | Prefix=2 | Prefix=3 |
|---|---|---|---|
| Unseen 4-byte | 0.0 | 39.5 | 60.5 |
| Unseen 3-byte (control) | 48.2 | 51.1 | 19.4 |

This pattern aligns with the intuition that more frequent exposure during training leads to better internalization of byte-level structure. Characters in the Common tier appear more often across diverse contexts, providing the model with more opportunities to learn the correspondence between semantic content and UTF-8 byte patterns. The monotonic decrease in validity from Common to Unseen tiers suggests that byte-sequence generation capability is fundamentally tied to training frequency, even when the characters themselves are out-of-vocabulary for the subword tokenizer.

The Unseen tier's 86.97% partial-credit validity rate, accompanied by the highest perplexity among all tiers, demonstrates meaningful zero-shot generalization to novel codepoints. The Unseen tier consists entirely of 4-byte UTF-8 characters (CJK Unified Ideographs Extension B and beyond, codepoints above U+10000), whereas the Common, Uncommon, and Rare tiers contain 3-byte characters from the Basic Multilingual Plane. This means the Unseen tier tests generalization to a different byte-pattern family: sequences beginning with 11110xxx (4-byte) rather than 1110xxxx (3-byte). The 86.97% partial-credit validity rate thus reflects the model's ability to generalize UTF-8 structural rules across byte-length boundaries, despite never having encountered these specific characters during training. Analysis of generated tokens indicates that the model often correctly identifies the need for a 4-byte sequence but struggles to select valid continuation bytes for start-byte combinations it has never encountered. This partial generalization suggests that UTF-8 structural learning involves both pattern-specific memorization from training exposure and limited abstract rule induction.

### 5.1.2. BYTE-LENGTH VS. FREQUENCY

The Unseen tier in the main Level 0 set is dominated by 4-byte CJK Extension B characters, which confounds two factors: novelty (the model has never seen the character) and byte-length class (the model has seen few 4-byte sequences overall). To separate them, we constructed a control set of 139 unseen 3-byte CJK ideographs (exhaustive over the unseen 3-byte BMP characters in $U \setminus K$) and evaluated 299 checkpoints under the same protocol. Table 2 reports partial-credit validity at prefix lengths 1–3 for the 4-byte Unseen characters and the 3-byte control.

At prefix=1 the gap is 48.2 pp: the model continues 3-byte sequences from the familiar 1110xxxx lead byte at

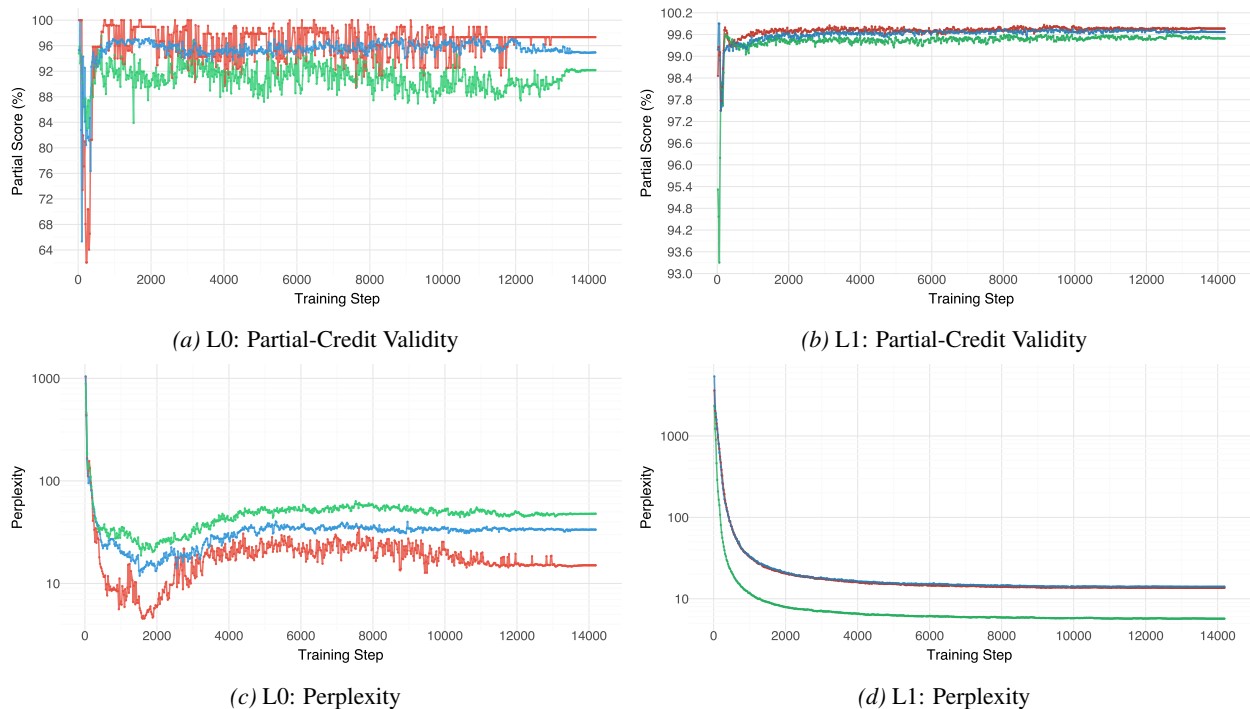

*(a)* L0: Partial-Credit Validity

*(b)* L1: Partial-Credit Validity

*(c)* L0: Perplexity

*(d)* L1: Perplexity

*Figure 4.* Side-by-side comparison of learning dynamics. The **left column** shows the baseline (L0) and the **right column** shows the context-guided setting (L1). Results of Chinese, Japanese, and Korean are plotted in green, red, and blue, respectively. Note how the partial credit validity (top row) stabilize significantly faster in the L1 setting.

48.2% validity, but produces 0% valid UTF-8 from the novel `11110xxx` lead. Inspecting outputs at prefix=1 for 4-byte targets, all 43 samples emit the identical bytes `F0 9F 92 95` (U+1F495), the sole 4-byte character the model encountered during training. The model has learned the 4-byte lead structure from this single example but collapses to one template; it never falls back to a 3-byte lead.

Within the 3-byte class, frequency has little effect; final-checkpoint partial-credit validity ranges only from 0.878 to 0.894 across Common, Uncommon, Rare, and Unseen-3B groups. The clear outlier is 4-byte Unseen (0.840). This indicates that byte-length dominates failures at this scale.

### 5.1.3. RELATIONSHIP WITH LANGUAGE MODELING

Figure 4 compares the evolution of partial-credit validity during training against that of perplexity and log-likelihood differences. We observe that while perplexity quickly converges to a low level, partial-credit validity converges more slowly: For partial-credit validity, convergence is observed after 740 training steps, while that for perplexity is observed as early as 380 training steps. For log-likelihood differences, the tendency is similar to that of the perplexity. We thus conclude that partial-credit validity does not correlate directly with either perplexity or log-likelihood relative to the gold standard. This suggests that our proposed metric captures a distinct aspect of a language model's ability to generate

correct byte sequences, making it a useful and important complement to existing measures.

### 5.2. Level 1: Context-Guided Evaluation

Level 1 evaluation mirrors typical language model usage, where the model generates text within semantic and syntactic context. Preceding words and phrases constrain plausible continuations, enabling the model to retrieve correct byte sequences by leveraging learned associations between concepts and their byte-level representations. Most practical generation scenarios provide such contextual cues.

#### 5.2.1. STRUCTURAL AND SEMANTIC EVALUATION

Structural validity and semantic correctness diverge. As in Figure 4, the model achieves high partial-credit validity in the context-guided setting, but Term Match Rate, i.e., whether it generates the specific correct character, reached 60.30% (Figure 5). The model has mastered UTF-8 encoding *mechanics* (generating valid byte sequences) but struggles with *semantics* (mapping context to the correct character).

The model often generates structurally valid characters that are semantically or phonetically related to the target but not exact matches. When prompted with context suggesting a particular kanji, the model may produce a different kanji

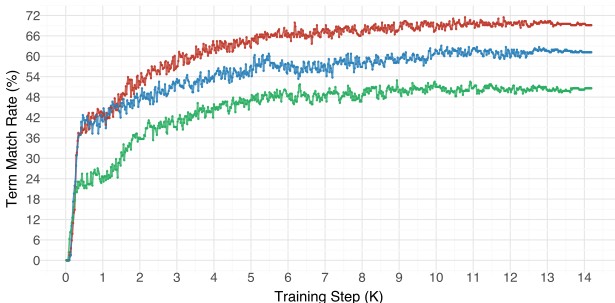

*Figure 5.* Term match rate. Results of Chinese, Japanese, and Korean are plotted in green, red, and blue, respectively.

with similar radical structure or meaning. Byte-level syntax and byte-level semantics are distinct capabilities, with the latter requiring more training.

The diagnostic $\Delta_{LL}$ (Eq. 5) splits the semantic failures further. Among Level 1 failures where $\Delta_{LL} > 0$ at the final checkpoint (51 cases), the model assigns higher teacher-forced likelihood to the gold continuation but greedy decoding emits a different byte. In 100% of these cases the emitted byte is a UTF-8 continuation byte in 0x80–0xBF, and for each lead byte the emission is a single mode: 0xE3 → 0x80 (CJK Symbols), 0xEC → 0x97 (Common Hangul), 0xF0 → 0x9F (Emoji range). The failure is mode collapse in $P(\text{byte}_2 \mid \text{byte}_1)$: the model has learned the marginal argmax for each lead byte but does not condition on the target character. Because $\Delta_{LL} > 0$, beam search or temperature sampling may recover these cases without retraining; verifying this empirically requires a decoding ablation we leave to future work. Appendix L reports the full distractor distribution.

### 5.2.2. CONTEXT HELPS STRUCTURAL LEARNING

Comparing Level 1 with Level 0 on the Common tier shows faster partial-credit validity convergence in the context-guided setting (Figure 4). The model achieves reliable UTF-8 generation earlier when semantic context is provided, suggesting contextual associations facilitate byte sequence retrieval before the model has fully internalized UTF-8 structural rules in isolation.

Cross-language patterns in Level 1 parallel Level 0: Japanese characters achieve high validity earlier than Korean and Chinese. The gap between languages is narrower in Level 1, indicating that context partially compensates for sparser byte-pattern exposure in larger character inventories.

### 5.2.3. RELATIONSHIP WITH LANGUAGE MODELING

Figure 4 compares the evolution of partial-credit validity during training against that of perplexity and log-likelihood differences. The model generates a single token given a

2-byte prefix. Unlike Level 0 evaluation, we observe that in Level 1, partial-credit validity converges faster than perplexity and log-likelihood differences. This suggests that generating byte sequences with contextual guidance (Level 1) is easier than context-free generation (Level 0). Nevertheless, the observation that partial-credit validity converges faster than perplexity further underscores the need for a metric beyond perplexity to evaluate a model's ability to generate valid byte sequences, highlighting the importance of our proposed metrics.

The finding that partial-credit validity converges faster than perplexity in Level 1, while the opposite holds in Level 0, further underscores that these metrics capture fundamentally different aspects of model capability. Perplexity measures the model's uncertainty over the full continuation distribution, whereas UTF-8 validity measures only whether the generated sequence satisfies structural encoding constraints. The divergent convergence patterns across evaluation levels demonstrate that neither metric subsumes the other, validating the need for dedicated partial-credit validity evaluation in byte-level language models.

## 6. Related Work

Byte-level tokenization eliminates vocabulary bottlenecks and handles the full diversity of Unicode characters, particularly for morphologically rich and logographic languages where subword tokenization faces inherent limitations. Xue et al. (2022) introduced ByT5, demonstrating that byte-level models could match the performance of token-based models while offering improved noise robustness. Their work showed that operating directly on UTF-8 bytes eliminates the need for language-specific preprocessing and handles any Unicode input without information loss. More recently, Pagnoni et al. (2025) presented the Byte Latent Transformer (BLT), achieving comparable performance to Llama 3 while using 50% fewer inference FLOPs, suggesting byte-level architectures may offer computational advantages at scale.

The challenges of Unicode processing in LLMs have been documented across multiple dimensions. Rust et al. (2021) demonstrated that morphologically rich languages require significantly more tokens than English for equivalent semantic content, creating systematic biases in multilingual models. The long-tail distribution of characters exacerbates this problem. For example, (Singh et al., 2024) found that low-resource languages suffer from poor tokenization efficiency, leading to degraded performance that discourages their use in training data, creating a vicious cycle.

Recent security research has revealed how tokenization vulnerabilities can be exploited. Geh et al. (2025) discovered that LLMs retain semantic understanding of non-canonical tokenizations despite never encountering them during train-

ing, enabling attackers to bypass safety filters through alternative word segmentations. The "glitch token" phenomenon, analyzed by Li et al. (2024) and further investigated by Land & Bartolo (2024), identified tokens like "SolidGold-Magikarp" that cause unpredictable model behavior. These tokens cluster in the embedding space and result from insufficient training.

Prior art addresses invalid byte generation at decode time rather than at training time. Constrained decoding masks tokens whose continuations would violate a grammar or automaton, guaranteeing that the emitted sequence stays in a target language (Willard & Louf, 2023; Koo et al., 2024). Cognetta & Okazaki (2025) formalize BPE and WordPiece as finite-state transducers, enabling subword-aware pattern promotion that is consistent with the tokenizer. Encoding UTF-8 validity in the same framework removes structural failures by construction, since UTF-8 is a regular language over bytes (§ 3.2.1). Our work is complementary: we measure when byte-level models acquire UTF-8 competence during training and separate the structural failure mode addressed by constrained decoding from the semantic failure mode (Term Match, Sec. 3.2.2) that it leaves unchanged.

While Kaplan et al. (2020) and Hoffmann et al. (2022) established power-law relationships between model scale and performance, subsequent work has shown these relationships break down for multilingual and Unicode-sensitive tasks. Pokharel et al. (2025) demonstrated that in zero-shot multilingual scenarios, model scale has minimal effect on performance. This suggests that scaling alone cannot overcome the fundamental vocabulary bottleneck created by tokenization, motivating our investigation into the minimum scale required for reliable UTF-8 sequence generation. Our findings through the UTF-8 expression can be applied to systems using its alternatives proposed in recent years (Limisiewicz et al., 2024; Land & Arnett, 2025).

## 7. Discussion

Languages vary in UTF-8 validity learning rates. Japanese reaches reliable validity earlier than Korean and Chinese, but two evaluation-set properties contribute beyond character inventory size. The Japanese OOV set contains 36 characters (mostly Kana, since most Kana are vocabulary-covered) compared with 256 for Korean and 256 for Chinese, which biases the language-level average upward. Kana also occupy a narrow Unicode range (U+3040–U+30FF) with regular E3 8x xx byte patterns, whereas Hangul (lead bytes EA–ED) and CJK ideographs (E4–E9) span a much larger byte-pattern space. The faster Japanese convergence reflects these structural and sampling properties, not a language-specific capability. The Unseen tier's 86.97% validity on 4-byte characters never encountered during training reflects generalization across byte-length boundaries, with the byte-

*Table 3.* Cross-model evaluation at Level 0/1. $V_p$: partial-credit validity, $V_s$: strict validity, $M$: term match rate. All values (%) at generation step 5, averaged across languages and prefix lengths.

| Family | Size | $V_p$ | $V_s$ | Gap | L1 $M$ |
|---|---|---|---|---|---|
| Baseline | 0.4B | 93.5 | 39.8 | 53.6 | 47.8 |
| Gemma-3 | 1B | 33.6 | 0.9 | 32.6 | 0.3 |
| Gemma-3 | 4B | 100 | 35.7 | 64.3 | 0.0 |
| Llama-3.2 | 1B | 96.9 | 38.4 | 58.5 | 0.8 |
| Llama-3.2 | 3B | 97.8 | 49.0 | 48.8 | 1.0 |
| Llama-2 | 7B | 98.4 | 33.9 | 64.5 | 33.1 |
| Mistral | 7B | 96.0 | 49.9 | 46.1 | 23.3 |
| OLMo-2 | 1B | 93.0 | 33.5 | 59.5 | 7.0 |
| OLMo-2 | 7B | 93.5 | 28.0 | 65.4 | 14.3 |
| Qwen-3.5 | 4B | 99.2 | 87.0 | 12.2 | 2.4 |
| Qwen-3.5 | 9B | 99.7 | 89.8 | 9.9 | 0.5 |

length dimension identified as the dominant axis in our control experiment (Sec. 5.1.2).

Our findings are not an argument against smaller models, which remain essential for edge devices, real-time systems, and resource-constrained environments. Smaller models serve as research tools, distillation targets, and domain-specific solutions. The findings instead clarify training requirements for byte-level tokenization and metrics to monitor for reliable UTF-8 generation.

### 7.1. Cross-Model Validation

To test whether the structural-semantic gap generalizes beyond our 355M baseline, we evaluated 10 open-weight models from 5 families (1B–9B parameters) using the same Level 0 and Level 1 protocol and evaluation data. Results are summarized in Table 3; checkpoint sources, tokenizer handling, and decoding settings are reported in Appendix K.

The $V_p$–$V_s$ gap persists across all models and scales tested (9.9–65.4 pp). Gap magnitude is stable within families, where OLMo-2 shows 60–65 pp gaps at both 1B and 7B; Qwen-3.5 shows 10–12 pp at both 4B and 9B, but varies substantially across families. $V_s$ generally improves with scale (Llama-3.2: +10.6 pp from 1B to 3B; Qwen-3.5: +2.8 pp from 4B to 9B), but Level 1 term match does not improve consistently. This confirms that structural validity and semantic correctness are distinct: larger models produce more valid byte sequences, but often not the *correct* character.

We cannot measure convergence rates on the open models without checkpoint-level sweeps, but cross-model evidence is consistent with the convergence-lag hypothesis from our baseline. If $V_s$ caught up to $V_p$ given enough training, we would expect smaller gaps in larger models that have seen more tokens, and the data does not show this trend.

Tokenizer design is a stronger predictor of semantic byte completion than model size. SentencePiece byte-fallback models (baseline at 47.8%, Llama-2 at 33.1%, Mistral at 23.3%) achieve higher term match than GPT-2-style BPE models with larger vocabularies (Qwen-3.5 9B at 0.5%, Gemma-3 4B at 0.0%). Small vocabularies force frequent byte-fallback during training, giving the model more practice at byte-level generation.

## 7.2. Limitations and Future Work

Our training dynamics analysis (the convergence gap between perplexity and validity) is conducted at a single 355M-parameter training run; "scale-dependent" here refers to training-token scale under this setup rather than a scaling law. The cross-model evaluation confirms that the structural-semantic gap exists at larger scales, but we do not have checkpoint-level sweeps for open models to measure convergence *rates* at those scales. We also emphasize that $V_{partial}$ is a diagnostic; deployed applications typically require fully decodable strings, for which $V_{strict}$ is the operational metric.

Our Level 1 contexts are generated by Gemini 3 Pro and filtered with formal criteria (language identification, uniqueness, length); we do not apply human semantic validation. Context naturalness, target-character appropriateness, and synthetic-generator bias can therefore affect Term Match measurements, and the synthetic distribution may differ from naturally occurring text.

Our experiments focus on East Asian languages (Chinese, Japanese, Korean) plus English. These languages were chosen for their diverse character sets and multi-byte encoding requirements, but findings may not replicate to other languages. The evaluation framework itself is language-agnostic and applicable to any script encoded in UTF-8.

Constrained decoding methods (Koo et al., 2024; Willard & Louf, 2023; Cognetta & Okazaki, 2025) can guarantee $V_{strict} = 1.0$ by masking structurally invalid tokens at each generation step. However, our distractor analysis (Appendix L) shows that in 100% of $\Delta_{LL} > 0$ failure cases, the model already produces structurally valid continuations: the failures are semantic, not structural. Constrained decoding addresses structural failures, but not semantic ones. Since $\Delta_{LL} > 0$ implies the model assigns higher probability to the gold completion under teacher forcing, beam search or temperature sampling may recover the corresponding cases without retraining, though we leave a controlled decoding ablation to future work. The constrained-decoding intervention rate may serve as a training diagnostic, warranting further study.

A related question is whether the validity lag is specific to byte-level generation or a special case of slower learning on rare tokens. The two-fold lag we report compares byte-level validity against overall perplexity, which is dominated by common tokens, so part of the gap is the standard long-tail versus common-case gap. Two observations argue for distinct dynamics. First, the failure mode is concentrated mode collapse rather than distributed uncertainty: in 100% of $\Delta_{LL} > 0$ cases, each lead byte maps to a single continuation byte (Appendix L). Generic rare-token failures show a spread of plausible alternatives. Second, tokenizer setup overrides scale in the cross-model evaluation: SentencePiece byte-fallback models with small vocabularies outperform much larger models with large vocabularies on term match. Cleanly separating byte-level dynamics from rare-subword dynamics would require two ablations we leave for future work: tracking per-cohort cross-entropy on subword tokens versus byte-fallback sequences sampled at matched training frequencies, and training two otherwise-identical models that differ only in CJK vocabulary coverage.

Architectural changes such as explicit byte-position encodings or hierarchical representations may accelerate UTF-8 learning. Context-rich generation achieves reliable UTF-8 output at lower training scales than context-sparse generation, though structural validity does not guarantee semantic correctness, and the tradeoff needs further investigation.

# 8. Conclusion

UTF-8 validity is a distinct capability that emerges at different rates than standard language modeling metrics. In experiments with a 355M parameter model evaluated across 420 checkpoints, UTF-8 validity convergence lagged perplexity convergence, perplexity stabilizes after approximately 2.1B training tokens, while UTF-8 validity requires roughly 4.2B tokens, a two-fold difference. Practitioners deploying byte-level models should note that a model appearing well-trained by perplexity standards may still produce invalid UTF-8 sequences.

Character frequency correlates with structural validity (Common: 96.21%, Uncommon: 95.57%, Rare: 95.26%), but a control experiment indicates that byte-length exposure is the dominant axis of failure at this scale, with frequency contributing only modestly within a fixed byte-length class (Sec. 5.1.2). The gap between structural validity and semantic correctness remains stark: despite moderate validity rates, Term Match Rate reached 60.30%. Languages with larger character inventories (e.g., Korean and Chinese) need more training exposure than Japanese to reach comparable validity.

Context-guided evaluation shows that semantic context accelerates structural validity convergence, though semantic correctness remains low even with context. Training data exposure should exceed perplexity convergence thresholds, and UTF-8 validity should be monitored as a distinct metric.

## Acknowledgements

This work was supported by JSPS KAKENHI Grant Number 25H01137, the "R&D Hub Aimed at Ensuring Transparency and Reliability of Generative AI Models" project of the Ministry of Education, Culture, Sports, Science and Technology, and JST K Program Japan Grant Number JP-MJKP24C3.

## Impact Statement

This paper presents work whose goal is to advance the field of Machine Learning. There are many potential societal consequences of our work, none which we feel must be specifically highlighted here.

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

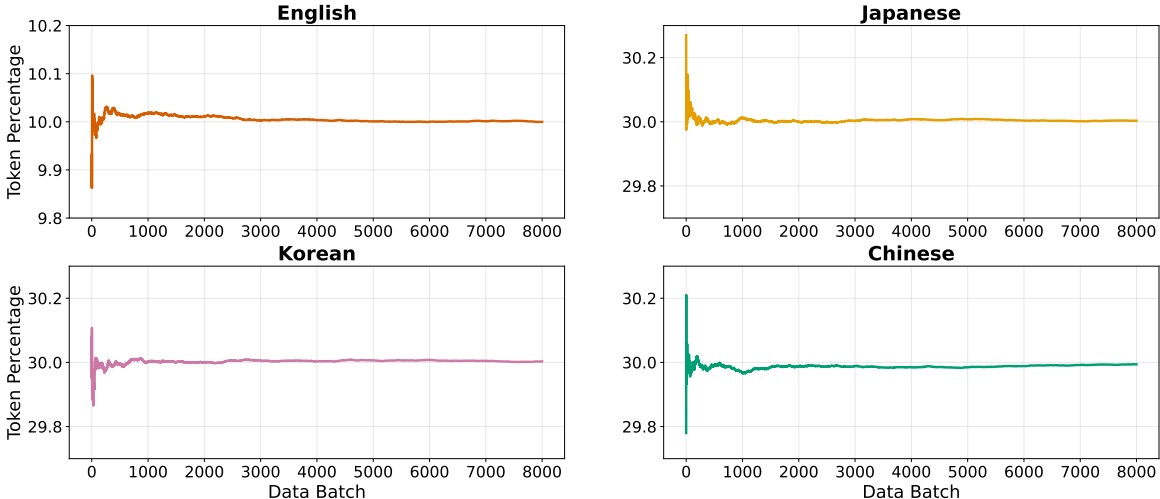

*Figure 6.* Token distribution convergence during training data construction. The adaptive weight adjustment algorithm dynamically modifies language sampling probabilities to achieve the target distribution while preserving natural sentence boundaries.

## A. Ethical Considerations

This research investigates potential vulnerabilities in language model tokenization that could be exploited for adversarial purposes. However, we believe the benefits of understanding these limitations outweigh the risks. Our work aims to characterize vulnerabilities so they can be addressed in future model architectures, ultimately improving the robustness of deployed systems. We do not introduce new attack methodologies but rather study the scaling properties of known tokenization issues. All adversarial sequences tested are synthetically generated rather than optimized for maximum harm. Our research provides concrete guidance on training requirements for reliable byte-level generation, helping practitioners make informed decisions about when byte-level tokenization is viable for their applications.

We have deliberately avoided testing our methods on production models or developing tools that could facilitate malicious use. All experiments are conducted on models we train ourselves, ensuring no impact on deployed systems. We find no significant ethical risks associated with this work beyond the general considerations of academic research.

## B. Model Architecture and Compute Budget

The GPT-2 variant model used for our work employs several standard modernization choices for training stability and efficiency. We replace LayerNorm with RMSNorm (Zhang & Sennrich, 2019), use Rotary Position Embeddings (RoPE) (Su et al., 2024), and adopt Grouped-Query Attention (GQA) (**?**) with a reduced number of key/value heads to lower memory usage (similar in spirit to recent implementations such as Gemma 2 (Gemma Team et al., 2024)). The feed-forward blocks use a gated MLP (GeGLU variant) (Shazeer, 2020). We further apply query scaling by $d_{\text{head}}^{-0.5}$ and embedding scaling by $\sqrt{d_{\text{hidden}}}$.

Training was conducted on a single node with 8 Nvidia B200 GPUs for one epoch over 76 hours. Evaluation used various single-accelerator setups across A6000s, RTX PRO 6000s, and an M4 Pro Mac Mini.

## C. Training Corpus Sampling Method Ablations

The main paper describes Weighted Dynamic sampling as the chosen method for training corpus construction. This appendix documents the alternative methods that were evaluated and rejected. Weighted Dynamic sampling was chosen because alternative methods either truncate sentences mid-thought (Strict Quota) or produce unnatural cross-lingual mixing within batches (Buffer-Balanced).

## C.1. Strict Quota

The Strict Quota method enforces exact token distribution by maintaining running quotas for each language, defined as follows:

$$Q_l^{t+1} = \max(0, Q_l^t - |d_l|) \tag{9}$$

$$P(l|Q) = \begin{cases} 1 & \text{if } Q_l > 0 \text{ and } Q_l = \max_k Q_k \\ 0 & \text{otherwise} \end{cases} \tag{10}$$

where $Q_l^t$ is the remaining quota for language $l$ at time $t$, and $|d_l|$ is the token count of document $d$ from language $l$. This method was rejected because it frequently truncates sentences mid-thought when quotas are exhausted, producing training examples that do not reflect natural language boundaries and potentially teaching the model to generate incomplete utterances.

## C.2. Buffer-Balanced

The Buffer-Balanced method maintains separate token buffers for each language and extracts proportionally from each buffer, defined as follows:

$$B_l^{t+1} = B_l^t \cup \text{tokens}(d_l) - E_l^t \tag{11}$$

$$E_l^t = \text{extract}(B_l^t, n_l) \tag{12}$$

$$n_l = \lfloor p_l \cdot S \rfloor \tag{13}$$

where $B_l^t$ is the buffer for language $l$, $E_l^t$ is the extracted tokens, $p_l$ is the target proportion, and $S$ is the block size.

## C.3. Qualitative Comparison

Table 4 shows detokenized text from 256-token training batches for each method.

## C.4. Quantitative Results

The analysis reveals a fundamental trade-off between distribution precision and text quality. Strict Quota achieves near-perfect distribution (0.44% MAE) with varying byte-fallback rates per language. Buffer-Balanced shows moderate distribution accuracy (1.36% MAE) with intermediate byte-fallback rates. Weighted Dynamic, despite higher distribution error (2.70% MAE), maintains the lowest byte-fallback rates across all languages, particularly for CJK languages.

# D. Language-Specific Accuracy and Byte-Fallback Statistics

Here, we report two per-language measurements taken on the training corpus. The first quantifies how closely each sampling method matches the target language proportions in the assembled corpus, complementing the aggregate MAE/RMSE numbers in Appendix C. The second quantifies how much of the resulting training signal arrives through byte-fallback tokens rather than full subword tokens; this is the quantity most directly relevant to byte-level learning dynamics studied in the main paper.

Table 6 reports the realized per-language token share across 20 trials of each sampling method. Strict Quota achieves the lowest deviation (as expected, since it enforces quotas by construction), at the cost of mid-sentence truncations documented in Appendix C. Buffer-Balanced and Weighted Dynamic both keep deviations within a few percentage points of target; we adopt Weighted Dynamic for the main training run because it preserves sentence boundaries while still tracking the target distribution.

Table 7 reports the byte-fallback rate per language, i.e., the fraction of training tokens emitted as raw byte tokens rather than subword tokens within each language partition. Chinese drives most of the byte-fallback load: at 30% of the corpus

*Table 4.* Example 256-token training batches from each sampling method. Strict Quota enforces exact distribution but truncates mid-sentence. Buffer-Balanced mixes languages unnaturally within blocks. Weighted Dynamic preserves sentence boundaries at the cost of distribution precision.

| Method | Example Batch Content |
|---|---|
| Strict Quota | **[EN: 26 tok]** The quick brown fox jumps over the lazy dog while... 
 **[KO: 77 tok]** 오늘은 날씨가 매우 좋아서 공원에 산책을 나갔습니다. 봄꽃들이 활짝 피어있고 새들이 노래를 부르고 있었어요. 많은 사람들이 가족과 함께 즐거운 시간을 
 **[JA: 77 tok]** 今日は天気がとても良かったので、公園に散歩に行きました。春の花が満開で、鳥たちが歌っていました。多くの人々が家族と一緒に楽しい時間を過ごしていました。でも急に雨が降り始めて、みんな慌てて 
 **[ZH: 76 tok]** 今天天气非常好，所以我去公园散步了。春天的花朵盛开着，鸟儿在唱歌。许多人和家人一起度过愉快的时光。但是突然开始下雨了，大家都慌慌张张地跑回家。我也赶紧找了个地方避雨，等雨停了才 
 *Text cuts off mid-sentence when quota is reached.* |
| Buffer-Balanced | The implementation follows a simple 프로그래밍 패턴을 사용하여 pattern where each コンポーネントは独立して component operates 独立运行并且可以 independently and can be easily replaced. 이러한 방식은 유지보수를 쉽게 만들고 メンテナンスが簡単になり、系统的灵活性也大大提高了。 
 *Unnatural language mixing within single training blocks.* |
| Weighted Dynamic | **[EN: 19 tok]** Machine learning has revolutionized how we process data. 
 **[KO: 82 tok]** 기계 학습은 우리가 데이터를 처리하는 방식에 혁명을 일으켰습니다. 특히 딥러닝 기술의 발전으로 이미지 인식, 자연어 처리, 음성 인식 등 다양한 분야에서 놀라운 성과를 거두고 있습니다. 
 **[JA: 71 tok]** 機械学習は私たちのデータ処理方法に革命をもたらしました。特にディープラーニング技術の発展により、画像認識、自然言語処理、音声認識など様々な分野で驚くべき成果を上げています。 
 **[ZH: 84 tok]** 机器学习彻底改变了我们处理数据的方式。特别是深度学习技术的发展，在图像识别、自然语言处理、语音识别等各个领域都取得了令人惊叹的成果。这些技术正在改变我们的日常生活。 
 *Natural sentence boundaries preserved; distribution less precise.* |

*Table 5.* Performance metrics across 20 trials (100 blocks each)

| Method | MAE (%) | RMSE (%) | Time (ms) |
|---|---|---|---|
| Strict Quota | 0.44 | 0.52 | 85.5 |
| Buffer-Balanced | 1.36 | 1.60 | 96.6 |
| Weighted Dynamic | 2.70 | 3.12 | 76.0 |

with a 56.1% byte-fallback rate, it contributes roughly $0.30 \times 0.561 / 0.227 \approx 74\%$ of all byte-fallback tokens seen during training. English is effectively absent from the byte-fallback signal, and Japanese/Korean contribute moderately. These per-language rates explain why the model has markedly more exposure to 3-byte CJK ideograph patterns than to other multi-byte structures and provide quantitative grounding for the byte-length effects analyzed in Section 5.1.2.

# E. UTF-8 DFA Transition Details

The UTF-8 validation DFA (Figure 2 in the main text) implements the full UTF-8 specification with explicit rejection of malformed sequences. The automaton has eight states: $S_0$ (initial/accepting), $S_1$ (awaiting 2-byte continuation), $S_2$ and $S_{2,1}$ (3-byte sequence states), $S_3$, $S_{3,1}$, and $S_{3,2}$ (4-byte sequence states), and $S_{\text{err}}$ (error sink).

Transitions from $S_0$ are determined by the lead byte:

- `00-7F`: ASCII, self-loop to $S_0$
- `C2-DF`: 2-byte lead, transition to $S_1$
- `E0-EF`: 3-byte lead, transition to $S_2$
- `F0-F4`: 4-byte lead, transition to $S_3$
- `80-BF`, `C0-C1`, `F5-FF`: invalid lead bytes, transition to $S_{\text{err}}$

*Table 6.* Language-specific distribution accuracy. Values are the realized percentage of training tokens in each language (mean $\pm$ s.d. across 20 trials of 100 blocks each).

| Language | Target | Strict Quota | Buffer-Balanced | Weighted Dynamic |
|---|---|---|---|---|
| EN | 10.0 | $10.01 \pm 0.36$ | $10.51 \pm 2.68$ | $10.57 \pm 4.65$ |
| KO | 30.0 | $29.69 \pm 0.57$ | $29.37 \pm 1.50$ | $28.80 \pm 2.55$ |
| JA | 30.0 | $30.18 \pm 0.57$ | $30.26 \pm 1.29$ | $29.98 \pm 2.80$ |
| ZH | 30.0 | $30.11 \pm 0.56$ | $29.85 \pm 0.89$ | $30.65 \pm 3.26$ |

*Table 7.* Per-language byte-fallback rates measured on the trained corpus under the Weighted Dynamic sampling configuration. The byte-fallback rate is the fraction of training tokens emitted as byte-fallback tokens within each language partition. Total tokens 45.47B; byte-fallback tokens 10.30B; non-byte tokens 35.16B.

| Language | Corpus share | Byte-fallback rate |
|---|---|---|
| English | 10% | 0.2% |
| Japanese | 30% | 11.1% |
| Korean | 30% | 8.3% |
| Chinese | 30% | 56.1% |
| Overall | 100% | 22.7% |

The `80-BF*` transitions in 3-byte and 4-byte paths indicate context-dependent continuation byte ranges that reject overlong encodings (e.g., `E0` requires `A0-BF` as the first continuation) and surrogate halves (`ED` requires `80-9F`). Similarly, `F0` requires `90-BF` and `F4` requires `80-8F` to reject codepoints above U+10FFFF.

## F. Level 0 Trial Set Construction

The Level 0 trial set evaluates context-free UTF-8 generation by prompting the model with isolated OOV characters under byte-fallback. We reuse the frequency-tiered dataset described in Section I, stratifying by *Common*, *Uncommon*, *Rare*, and *Unseen* tiers.

For each character $c$ in the trial set, we construct the prompt by tokenizing $c$ (which produces byte-fallback tokens) and providing the first $k$ bytes as context. The model must generate the remaining bytes to complete a valid UTF-8 character. We sweep $k \in \{1, 2\}$ for 3-byte characters and $k \in \{1, 2, 3\}$ for 4-byte characters to evaluate partial-completion difficulty.

## G. Level 1 Prompt Construction

Level 1 evaluates context-guided byte retrieval by embedding OOV characters in natural language sentences. Target characters are sourced from the pre-tokenized training corpus by identifying contiguous byte-fallback sequences (e.g., `<0xE4><0xB8><0xAD>`) and decoding them to UTF-8.

For each target character $c$ with byte sequence $B_c = (B_p, B_r)$, where $B_p$ is the provided prefix and $B_r$ is the suffix to generate, we construct a prompt $P = C_{\text{ctx}} \| B_p$ where $C_{\text{ctx}}$ is the preceding sentence context. The model is evaluated on whether it emits exactly $B_r$ as the immediate continuation.

We use a fixed split of 256 prompts per language (Japanese, Korean, Chinese) for checkpoint-wise monitoring, with prompts stratified by character frequency to ensure coverage across difficulty levels.

## H. Other Evaluation Metrics

At the final checkpoint (step 14,189, corresponding to 80B tokens), binary strict validity—which requires the entire generated sequence to be valid UTF-8—was highest on the *Common* tier (50.47%), followed by *Unseen* (33.33%), *Uncommon* (32.48%), and *Rare* (30.24%). The binary soft metric, which credits complete characters without penalizing trailing incomplete bytes, showed higher scores across all tiers: *Common* (92.37%), *Uncommon* (90.18%), *Rare* (89.67%), and *Unseen* (79.62%).

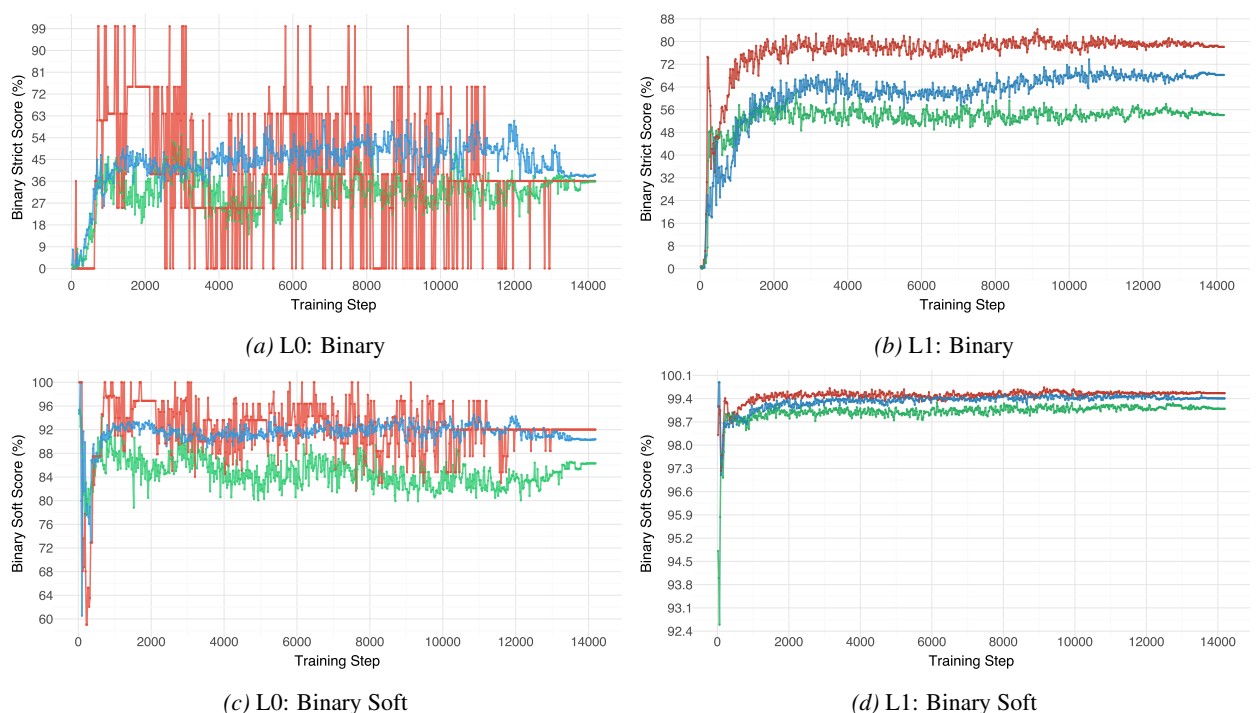

*(a)* L0: Binary

*(b)* L1: Binary

*(c)* L0: Binary Soft

*(d)* L1: Binary Soft

*Figure 7.* Side-by-side comparison of other validity metrics evaluated. The **left column** shows the baseline (L0) and the **right column** shows the context-guided setting (L1). Results of Chinese, Japanese, and Korean are plotted in green, red, and blue, respectively. Note how L0 in general is very unstable, while L1 is relatively more stable.

### H.1. Binary Score

The binary score awards credit only for completely valid sequences:

$$V_{\text{binary}}(B) = \begin{cases} 1 & \text{if DFA ends in } S_0 \text{ with no errors} \\ 0 & \text{otherwise} \end{cases} \tag{14}$$

This score is appropriate for final evaluation where partial progress is insufficient.

### H.2. Binary Soft Score

A third variant credits only complete characters without penalizing trailing incomplete bytes:

$$V_{\text{binary-soft}}(B) = \frac{b_c}{|B|} \tag{15}$$

This score distinguishes between invalid sequences (which reduce $b_c$) and merely incomplete ones (which do not).

## I. Level 0 and 1: Building the Evaluation Character Set

To evaluate the model's zero-shot generalization on out-of-vocabulary (OOV) characters, we constructed a stratified dataset $D_{trial}$ of 4,000 characters representing four frequency tiers within the training corpus: *Common*, *Uncommon*, *Rare*, and *Unseen*.

We defined the universe of known characters $K$ as the union $K = V \cup S$, where $V$ is the set of unique Unicode characters in the model's Byte-Pair Encoding (BPE) vocabulary tokens, and $S$ is the set of unique OOV characters in the training corpus as recorded in frequency snapshot $F$. This definition excludes any character the model encountered during training, either as a token or byte-fallback sequence. We obtained $|K| = 50{,}708$.

The dataset $D_{trial}$ comprises four disjoint subsets of $N = 1000$ samples each. We applied a balanced sampling strategy to

ensure linguistic diversity, targeting equal distribution of Han (CJK Ideographs), Hangul (Korean), and Kana (Japanese) scripts, plus other symbols. The natural distribution of OOV characters constrained this balance.

### I.1. Common Tier

The **Common Tier** ($T_{common}$) was selected from the OOV characters used in the Level 1 (Context-Guided) evaluation, enabling direct comparability between Level 0 and Level 1 performance on high-frequency characters. We prioritized the 244 characters from the Level 1 Trial set and filled the remainder with script-balanced selection from the Level 1 source data.

### I.2. Uncommon and Rare Tiers

For the **Uncommon Tier** ($T_{uncommon}$) and **Rare Tier** ($T_{rare}$), we employed script-aware sampling rather than global frequency thresholds to ensure linguistic diversity. The global distribution of OOV character frequencies is skewed, with CJK Ideographs dominating the long tail. To prevent mono-script tiers, we stratified selection by script type. For the Rare Tier, we selected the 300 lowest-frequency characters for each of Han and Hangul, and 100 for Other. For Kana, we included all 28 OOV characters from the corpus snapshot, as most Kana are vocabulary-covered. For the Uncommon Tier, we selected the next-lowest 300 characters for Han and Hangul from the remaining pool. This ensures evaluation of the rarest Hangul even though Hangul characters generally appear more frequently than rare Han characters.

### I.3. Unseen Tier

The **Unseen Tier** ($T_{unseen}$) tests zero-shot generalization. We defined candidate universe $U$ comprising all Unicode codepoints in the CJK Unified Ideographs (including Extensions A and B) and Hangul Syllables blocks. The pool of candidate unseen characters was $P_{unseen} = U \setminus K$, yielding 42,871 candidates never observed in training. Due to high coverage of Basic Multilingual Plane (BMP) characters in training, this pool consists predominantly of CJK Extension B characters (codepoints above U+10000), which require 4-byte UTF-8 encoding rather than the 3-byte encoding used by BMP characters in the other tiers. This means the Unseen tier tests generalization to a different UTF-8 byte-pattern family (11110xxx start bytes) in addition to novel character identity. The final sample contains 988 Han characters and 12 Hangul characters.

### I.4. Dataset Interleaving

The final dataset $D_{trial}$ was constructed using deterministic interleaving across the four tiers. The sequence follows the repeating pattern $[T_{common}, T_{uncommon}, T_{rare}, T_{unseen}, \dots]$. The interleaved structure guarantees balanced coverage across frequency tiers for each language.

**Note on computational limitations.** Due to computational resource constraints, the experiments reported in the main text use a subset of $M = 256$ samples per language group (64 per frequency tier). A full sweep over the complete 4,000-character dataset is left for future work.

## J. Level 1: Synthetic Data Generation

To avoid data leakage from re-using pre-training text, we generate synthetic sentence contexts using Gemini 3 Pro. For each target OOV character $c$ and language $L$, we prompt the model:

> *Write a single grammatically correct sentence in [L] that naturally incorporates the character "[c]". The sentence should be 10–20 words and provide clear semantic context for the character.*

Generated sentences are filtered for: (1) presence of the target character, (2) correct language identification via langdetect, (3) uniqueness (no duplicate sentences across the dataset), and (4) length constraints (10–30 tokens after BPE tokenization). Approximately 15% of generated sentences are rejected by these filters.

## K. Cross-Model Evaluation Details

The cross-model results in Section 3 use publicly released instruction-tuned checkpoints retrieved from the Hugging Face Hub. We use the default tokenizer shipped with each checkpoint and rely on the model's native byte handling: SentencePiece-based models (Llama-2, Llama-3.2, Mistral, our baseline) expose explicit byte-fallback tokens, whereas GPT-2-style BPE models (Gemma-3, OLMo-2, Qwen-3.5) represent arbitrary bytes through their pre-tokenizer without dedicated byte-fallback tokens. For every model we feed the *same* UTF-8 byte prefixes derived from the Level 0 and Level 1 trial sets; the model is then asked to continue from the corresponding tokenization of that prefix. We use greedy decoding throughout, generate up to 5 tokens per trial, and average all reported values across languages and prefix lengths. All 100 trial samples per language are reused unchanged from the baseline protocol (Appendices F and G). Decoded byte streams are scored by the same UTF-8 DFA used for the baseline; partial-credit validity, strict validity, and Term Match are computed identically.

## L. Distractor Token Distribution at $\Delta_{LL} > 0$

At the final checkpoint, Level 1 has 51 cases where $\Delta_{LL} > 0$ (the model assigns higher teacher-forced likelihood to the gold continuation) but greedy decoding emits a different byte. All 51 emitted tokens are byte tokens in the range 0x80–0xBF; no subword token and no lead byte appears. Table 8 groups the emissions by lead byte, and Table 9 reports the full distribution of emitted bytes.

*Table 8.* For three lead bytes the model emits a single continuation byte across all $\Delta_{LL} > 0$ failures. Mode collapse is exact within each lead byte.

| Lead byte | Emitted (count) | Produces |
|---|---|---|
| 0xE3 (JA) | 0x80 (8/8) | CJK Symbols U+3000 |
| 0xEC (KO) | 0x97 (4/4) | Common Hangul |
| 0xF0 (4B) | 0x9F (3/3) | Emoji range |

*Table 9.* Full distribution of emitted bytes for the 51 $\Delta_{LL} > 0$ failures at the final checkpoint. All values lie in 0x80–0xBF (UTF-8 continuation byte range), so the structural-validity DFA accepts every emission. The semantic failure is a mode collapse in $P(\text{byte}_2 \mid \text{byte}_1)$.

| Emitted byte | Count |
|---|---|
| 0x80 | 12 |
| 0xB7 | 5 |
| 0x97 | 4 |
| 0x8B | 4 |
| 0x8C | 3 |
| 0x9F | 3 |
| 0xB9 | 2 |
| Other (18 singletons) | 18 |
| Total | 51 |

