# OpenReview forum: "Beyond Perplexity: UTF-8 Validity in Byte-aware Language Models"
_ICML.cc/2026/Conference — ICML 2026 regular_

### Official Review · Reviewer_TrWg · 2026-02-22

**Soundness:** 3
**Presentation:** 3
**Significance:** 3
**Originality:** 3
**Overall Recommendation:** 4
**Confidence:** 3

**Summary:**

The paper highlights a practical blind spot in byte-level LMs: despite “universal” byte coverage, generation can yield invalid UTF-8 (e.g., broken multibyte characters), and perplexity does not reliably reflect this risk. The authors introduce a focused UTF-8 evaluation suite—Level 0 (structural completion without context) and Level 1 (context-guided byte completion)—with a DFA-based partial-credit validity metric (plus complementary diagnostics), enabling analysis of UTF-8 validity and revealing it can lag perplexity during training.

**Compliance With Llm Reviewing Policy:**

Affirmed.

**Final Justification:**

The rebuttal addressed my main concerns

**Key Questions For Authors:**

1. Would constrained decoding improve the UTF-8 validity?

**Limitations:**

yes

**Strengths And Weaknesses:**

**Strengths**
1. This paper discovers an under-explored failure mode: Byte-level LMs can generate invalid UTF-8 for rare/unseen characters, which can break downstream decoding and text generation reliability.
2. Clear empirical takeaway beyond perplexity: The paper demonstrates UTF-8 validity “converges” much later than perplexity, so perplexity alone can be misleading for deployment readiness.

**Weaknesses**
1. Limited Model Scale: The experiments were conducted on a 355M parameter model. While this is great for edge-device research, the authors admit they don't know if the perplexity-validity gap persists in massive

---

> ### Author Rebuttal · Authors · 2026-03-31
>
> ## Weakness 1. While this is great for edge-device research, the authors admit they don't know if the perplexity-validity gap persists in massive…
>
> We have conducted further experiments in response to a similar concern, and the results are as follows:
>
> | Family | Size | L0 $V_p$ | L0 $V_s$ | Gap | L1 $M$ |
> |---|---|---|---|---|---|
> | Baseline | 0.4B | 93.5 | 39.8 | 53.6 | 47.8 |
> | Gemma-3 | 1B | 33.4 | 1.0 | 32.4 | 0.2 |
> | Gemma-3 | 4B | 100 | 35.3 | 64.7 | 0.0 |
> | Llama-3.2 | 1B | 96.9 | 38.4 | 58.5 | 0.8 |
> | Llama-3.2 | 3B | 97.8 | 49.0 | 48.8 | 1.0 |
> | Llama-2 | 7B | 98.4 | 33.9 | 64.5 | 33.1 |
> | Mistral | 7B | 96.0 | 49.9 | 46.1 | 23.3 |
> | OLMo-2 | 1B | 93.0 | 33.5 | 59.5 | 7.0 |
> | OLMo-2 | 7B | 93.5 | 28.0 | 65.4 | 14.3 |
> | Qwen-3.5 | 4B | 99.2 | 87.0 | 12.2 | 2.4 |
> | Qwen-3.5 | 9B | 99.7 | 89.8 | 9.9 | 0.5 |
>
> The gap persists across all 10 open-weight models tested (9.9–65.4pp), with gap magnitude varying more across model families than across scale within a family. Full analysis is provided in our response to Reviewer eCfP (We figured copying and pasting the same content makes the response unnecessarily noisy, do let us know if we should replicate that here).
>
> While we do not know if the perplexity validity gap trend exists in terms of training dynamics as we have not replicated a full training run, validating against the final model and observing the results, we do expect to see utility of our method even in larger training regimes.
>
> ## Question 1. Would constrained decoding improve UTF-8 validity?
>
> Almost certainly yes for structural validity. Our UTF-8 DFA has 8 states with deterministic transitions (Figure 2 in paper). At each generation step, the DFA state identifies which byte values are valid continuations; all others can be masked out and the distribution renormalised. This guarantees $V_{\text{strict}}=1.0$ by construction, and has already been investigated by Koo et al. (2024) and Cognetta and Okazaki (2025). The intervention rate (how often the constraint overrides the top prediction) should also provide a complementary training diagnostic, since interventions should be inversely correlated to model maturity. It could also track UTF-8 internalisation during training more smoothly than $V_{\text{strict}}$, which is binary per sample, and could enable curriculum-based
> approaches where the constraint is progressively relaxed.
>
> However, constrained decoding addresses only structural validity (validity), not semantic
> correctness (match rate). Analysing cases where $\Delta_{LL} > 0$ (model assigns higher
> likelihood to the correct completion under teacher forcing but greedy decoding fails), we find
> that in 100% of such cases the model already produces structurally valid continuation bytes - just the wrong ones. For example, after lead byte 0xE3 the model always emits 0x80 (producing CJK Symbols) rather than the target Kana character. These outputs already satisfy the DFA constraint, so constrained decoding would not change them. Term match rates would remain unaffected. This distinction between structural and semantic failure modes is central to our contribution: $V_{\text{partial}}$ and $V_{\text{strict}}$ measure structural validity, term match measures semantic correctness. Constrained decoding closes the structural gap but leaves the semantic gap untouched. We regret not mentioning this and relevant work, and will address this in CR upon acceptance.

---

> > ### Author Rebuttal · Reviewer_TrWg · 2026-04-02
> >
> > My concerns have been adequately addressed.

---

> > > ### Author Response · Authors · 2026-04-08
> > >
> > > Thank you for the favorable response, timely feedback, and the constructive discussion. This discussion has been very valuable for our work.

---

### Official Review · Reviewer_eCfP · 2026-03-12

**Soundness:** 2
**Presentation:** 3
**Significance:** 3
**Originality:** 3
**Overall Recommendation:** 3
**Confidence:** 4

**Summary:**

This paper investigates the ability of byte-aware language models to generate structurally valid UTF-8 sequences when dealing with out-of-vocabulary (OOV) characters via byte-fallback. The authors train a single 355M parameter GPT-2 style model on 80B tokens (English, Chinese, Japanese, Korean). They introduce an evaluation framework using a Deterministic Finite Automaton (DFA) to calculate partial-credit structural validity. Their empirical findings suggest that UTF-8 structural validity converges later in training than perplexity, and that providing semantic context significantly improves the model's ability to generate valid byte sequences.

**Compliance With Llm Reviewing Policy:**

Affirmed.

**Key Questions For Authors:**

1) *Scale dependency:* Probably the main question. Drawing general conclusions about convergence gaps ("a factor of two") from a single 355M model is empirically risky. Do you have any data at larger scales (e.g., 1B–3B parameters) to confirm that this is a fundamental property of byte-fallback tokenization rather than an artifact of this specific small capacity?

2) *Confounding variables in the Unseen tier:* The Unseen tier tests 4-byte UTF-8 sequences, whereas the other tiers test 3-byte sequences. How can you isolate the effect of a character being "unseen" from the increased structural complexity of generating a 4-byte sequence? Why weren't unseen 3-byte characters included as a control?

3) *Real-world applicability:* Have you evaluated your Level 0 and Level 1 prompt suites on any widely used, pre-trained open-weight models (e.g., Llama 3, Mistral, Qwen)? Demonstrating that frontier models also struggle with these specific UTF-8 DFA checks would vastly improve the paper's significance.

**Limitations:**

Yes. The authors explicitly acknowledge the computational limitations that restricted their experiments to the 355M parameter scale. However, they should also acknowledge the methodological confounder in their dataset design (mixing character frequency with byte-length complexity across tiers).

**Strengths And Weaknesses:**

**Soundness:**

- *Strengths:* The proposed DFA-based partial-credit metric (Eq. 1) is a mathematically sound, rigorous way to evaluate incomplete generation rather than relying on brittle binary checks.

- *Weaknesses:* The paper suffers from severe empirical limitations. First, the authors make broad claims about LLM training dynamics (e.g., "UTF-8 validity convergence lags perplexity by roughly a factor of two") based on a single 355M parameter model. In the current NLP landscape, training dynamics at 355M often do not extrapolate to 1B+ or 7B+ scales (and even more - 14B, 70B and more) due to phase transitions and shifting scaling laws. Second, there is a critical confounding variable in the experimental design of the "Unseen" tier. The Common/Uncommon/Rare tiers consist of 3-byte BMP characters (1110xxxx), while the Unseen tier consists of 4-byte characters (11110xxx). The significant drop in performance on the Unseen tier cannot be conclusively attributed to zero-shot generalization failure; it is equally likely that the model struggles because the 4-byte DFA path is geometrically longer and more complex.

**Presentation:**

- *Strengths:* The paper is exceptionally well-written, logically structured, and easy to follow. The distinction between structural validity, semantic correctness, and probabilistic preference is articulated very clearly.
- *Weaknesses:* Some minor typos (e.g., "validty" in the Figure 4 caption) are present but do not detract from the overall high quality of the writing.

**Significance:**

- *Strengths:* A pertinent problem considered by this manuscript is the hidden failure modes of byte-fallback tokenizers, which are heavily utilized in frontier models. This manuscript studies a relevant challenge regarding "glitch tokens" and Unicode generation stability.

- *Weaknesses:* The practical impact is severely bottlenecked by the lack of evaluation on existing, state-of-the-art open-weight models. If generating invalid UTF-8 is a pervasive issue, the authors should have applied their DFA metric to models like Llama-3-8B or Gemma-2-2B to demonstrate real-world vulnerability. Without this, the paper reads like a technical report on a custom toy model rather than a generalizable contribution to the field.

**Originality:**

- *Strengths:* Shifting the evaluation focus from standard language modeling metrics (NLL) to the structural mechanics of Unicode generation via DFA is a fresh and creative perspective.

---

> ### Author Rebuttal · Authors · 2026-03-31
>
> Thank you for your insightful review, and we understand your concerns — especially recognizing whether or not the method and patterns generalize with scale, along with a great call out on a suboptimal trade-off we made during the experiment setup. We have conducted extra experiments in the hope that the concerns are addressed and adds clarity to specific aspects of the paper that were lacking in rigor.
>
> ## Question 1. General conclusions about convergence gaps from a single 355M model is empirically risky
>
> We cannot show convergence dynamics at larger scale (we do not have checkpoint-level training sweeps for open models), but we can demonstrate that the structural-semantic gap exists across scales at inference time. We evaluated open-weight models using the same Level 0/Level 1 protocol and data as the paper:
>
> | Family | Size | L0 $V_p$ | L0 $V_s$ | Gap | L1 $M$ |
> |---|---|---|---|---|---|
> | Baseline | 0.4B | 93.5 | 39.8 | 53.6 | 47.8 |
> | Gemma-3 | 1B | 33.4 | 1.0 | 32.4 | 0.2 |
> | Gemma-3 | 4B | 100 | 35.3 | 64.7 | 0.0 |
> | Llama-3.2 | 1B | 96.9 | 38.4 | 58.5 | 0.8 |
> | Llama-3.2 | 3B | 97.8 | 49.0 | 48.8 | 1.0 |
> | Llama-2 | 7B | 98.4 | 33.9 | 64.5 | 33.1 |
> | Mistral | 7B | 96.0 | 49.9 | 46.1 | 23.3 |
> | OLMo-2 | 1B | 93.0 | 33.5 | 59.5 | 7.0 |
> | OLMo-2 | 7B | 93.5 | 28.0 | 65.4 | 14.3 |
> | Qwen-3.5 | 4B | 99.2 | 87.0 | 12.2 | 2.4 |
> | Qwen-3.5 | 9B | 99.7 | 89.8 | 9.9 | 0.5 |
>
> The $V_p$–$V_s$ gap persists at every scale (9.9–65.4pp). OLMo-2 maintains 60–65pp gaps at both 1B and 7B. The gap magnitude is stable within families across scale - OLMo ~60pp at both sizes, Qwen ~10pp at both sizes - but varies dramatically across families (10–65pp range). This indicates the gap is determined by architecture and training regime, and not simply resolved by scale. While we cannot directly measure convergence rates for these models, the persistence of the gap across fully-trained models of varying scale and training data size is consistent with the convergence-lag hypothesis, if validity eventually caught up to perplexity given sufficient training, we would expect smaller gaps in larger models trained on more tokens, and the data does not show this trend. $V_s$ generally improves with scale (Llama-3.2: +10.6pp, Qwen-3.5: +2.8pp), but term match does not: Llama-3.2 stays below 1%, Qwen-3.5 below 3% regardless of size. Larger models produce more structurally valid bytes but are no better at producing the correct character.
>
> We omit L1 $V_p$ and $V_s$ from the table as $V_p$ is above 99% for all models — context makes structural validity trivial — while $V_s$ varies but follows the same family-dependent pattern as L0.
>
> The baseline's high term match (47.8%) despite being the smallest model is notable: its 8K SP vocabulary forces frequent byte-fallback during training, giving the model extensive practice at byte generation. Models with larger vocabularies (100K–262K) tokenize most CJK characters as single tokens and rarely encounter byte sequences during training. The SP models at 7B (Llama-2 at 33.1%, Mistral at 23.3%) also rank highest among models, suggesting that tokenizer/data-driven byte exposure, not scale, is a factor for semantic completion.
>
> ## Question 2. The Unseen tier tests 4-byte UTF-8 sequences, whereas the other tiers test 3-byte sequences
>
> Mixing 3-byte and 4-byte was an oversight where we traded simplicity at the cost of clarity on our end, and we are very thankful that you pointed this out. Splitting provides a valuable insight, and upon acceptance, we will incorporate this into CR.
>
> | | Prefix=1 | Prefix=2 | Prefix=3 |
> |---|---|---|---|
> | Unseen 4-byte | 0.0% | 39.5% | 60.5% |
> | Unseen 3-byte (control) | **48.2%** | **51.1%** | 19.4% |
>
> We constructed a control of 139 unseen 3-byte CJK ideographs (exhaustive, not sampled) and evaluated over 299 checkpoints. At shared prefix lengths, the 3-byte control consistently outperforms 4-byte unseen: validity 48.2% vs 0.0% at prefix=1, $V_{\text{partial}}$ 0.941 vs 0.832 at prefix=2. Within the 3-byte class, frequency has no effect: Common (0.882), Uncommon (0.894), Rare (0.888), Unseen-3B (0.878) are within 1.6pp. The failure mode is at the byte-length boundary, not the frequency boundary - a character being "unseen" has negligible impact on structural validity as long as the byte-length class is familiar.
>
> ## Question 3. Level 0 and Level 1 prompt suites on any widely used, pre-trained open models?
>
> Yes, as disclosed above. Even the best (Qwen-3.5 9B, $V_s$=89.8%) retains a 9.9pp gap. Gemma-3 1B scores only 33% $V_p$. Gemma-3 4B achieves 100% $V_p$ but 0% term match (it emits whole-codepoint tokens rather than byte completions). These results demonstrate that the phenomena reported in the paper generalize: the structural-semantic dissociation, the $V_p$–$V_s$ gap, and the failure to complete unseen byte patterns are observed across different models, scaling up to 9B parameters.
>
> We hope this changes your thoughts on the utility and applicability of our work.

---

### Official Review · Reviewer_rY95 · 2026-03-13

**Soundness:** 4
**Presentation:** 4
**Significance:** 4
**Originality:** 3
**Overall Recommendation:** 5
**Confidence:** 4

**Summary:**

This paper addresses the tendency for byte-aware models to generate invalid UTF-8 sequences when dealing with rare or unseen characters. With byte-level tokenization, an LLM can handle any Unicode input if it learns the complex encoding rules. The study investigates, using a 355M-parameter Transformer trained on a balanced multilingual corpus (English, Japanese, Korean, and Chinese), how long it takes the model to learn to generate structurally valid UTF-8 sequences. Tracking UTF-8 validity across 420 checkpoints, the authors found that while perplexity stabilizes around 2.1B tokens, the model requires exposure to about twice as many (4.2B) tokens for UTF-8 validity to stabilize. The paper introduces two evaluation levels: Level 0 (context-free) and Level 1 (context-guided). It also proposes a "partial-credit" validity metric using a Deterministic Finite Automaton (DFA) to provide a more granular view of progress than a simple pass/fail binary score.

**Compliance With Llm Reviewing Policy:**

Affirmed.

**Final Justification:**

I believe the paper is strong (especially after the authors have resolved some of the questions I had before). I've improved my scores accordingly.

**Key Questions For Authors:**

- The "Unseen" tier consists mostly of 4-byte characters from CJK Extension B. Do you believe the 86.97% validity rate reflects a generalized understanding of the 4-byte pattern (11110xxx), or is the model relying on its learning from 3-byte patterns (1110xxxx)?

- Japanese characters achieved validity earlier than Chinese or Korean. What could be the reason for that?

- Regarding the $\Delta_{LL}$ metric: for the cases where $\Delta_{LL} > 0$ but greedy sampling failed, did you observe any specific "distractor" tokens that the model preferred over the correct byte completion?

**Limitations:**

Yes

**Strengths And Weaknesses:**

## Soundness

### Strengths

- The introduction of DFA-based partial-credit validity is a significant improvement over binary metrics, allowing for more stable monitoring across training steps.
- Evaluating 420 checkpoints across 80B tokens provides a high-resolution look at the learning dynamics.
- Separating structural validity (G1) from semantic correctness (G2) and probabilistic preference (G3) provides a nuanced understanding of model failure modes.

### Weaknesses

- As the authors acknowledge, the experiments are limited to a 355M parameter scale. It's not clear whether the "~2x lag" between perplexity and validity persists in larger (e.g. 7B-parameter) models.
- Focus largely on CJK languages. While appropriate for multi-byte testing, the findings might not fully generalize to scripts with different morphological or encoding properties.

## Presentation

The paper is well-structured, clear and easy to read.

## Significance

This work is significant for LLM developers. It highlights an important blind spot in standard evaluation: a model that looks well-trained based on perplexity may still occasionally produce broken text when faced with rare characters. As byte-level tokenization (e.g., ByT5, BLT) becomes more popular for its universality, understanding the minimum scale and data requirements for Unicode reliability is vital.

## Originality

Byte-level models have been studied (notably ByT5), but this paper is the first to quantify the convergence relationship between structural validity and standard language modeling loss. The hierarchical frequency stratification (Common vs. Unseen) provides a new way to measure how models generalize abstract encoding rules versus memorizing specific byte patterns.

---

> ### Author Rebuttal · Authors · 2026-03-31
>
> First of all, thank you for the favorable review and valuable insights. With this feedback we conducted a couple extra experiments using our method and hope that the findings from these experiments can add clarity to the questions asked. Thanks to the insightful feedback, we have some interesting findings worth sharing and will incorporate these results into CR upon acceptance.
>
> ## Weakness 1. Experiments limited to 355M, lag in larger model unproven
>
> Valid concern. We have tried to approximate this with larger open models (details in upcoming key question responses).
>
> ## Weakness 2. Focus largely on CJK languages
>
> This is a valid concern - there are long-tail languages which we also would have liked to include in scope, but two practical roadblocks made this challenging: 1) data availability and 2) external dependency and availability of human annotators qualitative analysis (CJK was possible between the authors). While only validated on CJK languages, our evaluation scheme is language agnostic, so the execution of this can be left as future work.
>
> ## Question 1. Does the 86.97% Unseen-tier validity reflect generalised understanding of the 4-byte pattern…
>
> The validity reflects structural generalisation of the 4-byte pattern, not reliance on 3-byte learning - but the model has only memorised one 4-byte character (emoji U+1F495). We confirm this through a control experiment. The Unseen tier is 98.7% 4-byte (CJK Extension B), which the training data contains none of. We isolated a set of 139 unseen 3-byte characters (also not seen in training) and compared different prefix lengths:
>
> | | Prefix=1 | Prefix=2 | Prefix=3 |
> |---|---|---|---|
> | Unseen 4-byte | 0.0% | 39.5% | 60.5% |
> | Unseen 3-byte (control) | **48.2%** | **51.1%** | 19.4% |
>
> The 0% at prefix=1 for 4-byte is an interesting finding - the model has never seen an 11110xxx lead byte outside of emoji and cannot produce valid completions from it alone. We also observe that at prefix=1 the gap is 48pp: the model continues 3-byte sequences from the familiar ‘1110xxxx‘ leader, but produces 0% valid UTF-8 from the novel ‘11110xxx‘. The crossover at prefix=3 is structural: prefix=3 for a 3-byte character gives the full character (model must generate arbitrary UTF-8, low constraint); for 4-byte, one continuation byte remains in a committed DFA state (high constraint, easier to satisfy).
>
> Examining generated bytes: all 43 unseen 4-byte samples at prefix=1 produce *identical* F0 9F 92 95 (U+1F495) which is the sole 4-byte training exemplar. The model does not fall back to 3-byte patterns: 100% of generated bytes are valid continuations (0x80--0xBF), never 3-byte leads. It has learned the ‘11110xxx‘ structure from emoji but collapses to one template.
>
> A frequency-tier analysis confirms frequency is not the relevant axis: within the 3-byte class, partial validity is 0.878-0.894 across Common through Unseen-3B. The only outlier is 4-byte Unseen (0.840), where the novel lead byte is the dominant difficulty.
>
> ## Question 2. Why did Japanese achieve validity earlier?
>
> Japanese has only 36 OOV evaluation characters (Kana) versus 256 each for Korean and Chinese. The smaller sample biases averages upward. Kana also occupy a narrow range (U+3040-U+30FF) with regular encodings (‘E3 8x xx‘), a smaller byte-pattern space than Hangul (lead EA-ED) or CJK ideographs (lead E4-E9). Faster convergence reflects evaluation-set properties and structural regularity, not a
> language-specific capability.
>
> ## Question 3. Did you observe specific distractor tokens when $\Delta_{LL} > 0$ but greedy decoding failed?
>
> Yes. All 51 such cases at the final checkpoint show mode collapse to the most frequent continuation byte for each lead byte. 100% are byte tokens, all in the 0x80--0xBF range:
>
> | Lead | Always emits | Produces |
> |---|---|---|
> | 0xE3 (JA) | 0x80 (8/8) | CJK Symbols U+3000 |
> | 0xEC (KO) | 0x97 (4/4) | Common Hangul |
> | 0xF0 (4B) | 0x9F (3/3) | Emoji range |
>
> The full distribution: 0x80 (12x), 0xB7 (5x), 0x97 (4x), 0x8B (4x), 0x8C (3x), 0x9F (3x), 0xB9 (2x), plus 18 singletons. No subword token or lead byte ever appears. This is mode collapse in $P(\text{byte}_2 \mid \text{byte}_1)$: the model learns the marginal argmax but cannot condition on the target character. Since $\Delta_{LL} > 0$, the model does assign
> higher probability to the correct completion under teacher forcing; beam search or sampling could recover these.

---

> > ### Author Rebuttal · Reviewer_rY95 · 2026-04-02
> >
> > Thanks for the answers. I keep my overall recommendation as "accept", but I've also raised my confidence and soundness scores.

---

> > > ### Author Response · Authors · 2026-04-08
> > >
> > > We would like to thank you for the constructive discussion and positive feedback. We sincerely appreciate your feedback and believe the discussion will result in a higher quality publication.

---

### Official Review · Reviewer_BL2Q · 2026-03-30

**Soundness:** 4
**Presentation:** 4
**Significance:** 1
**Originality:** 2
**Overall Recommendation:** 3
**Confidence:** 4

**Summary:**

This paper explores the extent to which a particular large language model trained with byte-level tokenization will sometimes generate invalid characters, in terms of UTF-8 byte encoding sequences, depending on how long it has been trained for, the context available, etc. Success in producing valid byte sequences is compared to measurements of perplexity, etc.

**Compliance With Llm Reviewing Policy:**

Affirmed.

**Final Justification:**

Given the existence of constrained decoding, I am still somewhat dubious of the importance of this problem, but I do more see its relevance based on the extra results on current models that the authors provided in their rebuttal, hence I have raised my evaluation one notch. But overall I think soundness and clarity are fine, but significance is still lacking.

**Key Questions For Authors:**

None

**Limitations:**

Yes

**Strengths And Weaknesses:**

# Strengths
This paper appears to be technically sound and valid and it is presented in a simple and understandable manner. It shows results on the imperfect gradual emergence of valid byte sequence generation in a small (GPT2-style, 355M parameter) language model.

# Weaknesses
While there are valid experiments here, overall I just don't feel like the problem is central, relevant, and deep enough, and in particular, deeply enough connected to machine learning theory or practice to justify acceptance at ICML. There are sound experiments, but not a lot of originality of methods or insights. This just isn't a key, deep problem animating the ML community.

A particular weakness is that there are now fairly well documented methods to avoid ever generating invalid byte sequences (or invalid tool calls, etc.) through the method of constrained decoding. It is not mentioned in the paper but sort of obviates any need to study the question of this paper. Relevant papers include:
- https://openreview.net/forum?id=BDBdblmyzY
- https://arxiv.org/abs/2310.07075

---

> ### Author Rebuttal · Authors · 2026-03-31
>
> Thank you for the review. We will try to add some clarity to the two points raised.
>
> ## Weakness 1. Value of the problem
>
> We understand the basis of your concerns - byte sequencing ability is a relatively small portion of a larger system and can be considered a minor area in the grand scheme of LLM training. However, we have evaluated 10 open-weight models from 5 families (1B–9B) using the same protocol as the paper and are happy to share some observations that we believe can be of interest to the greater community, as they highlight that byte sequencing isn’t necessarily a solved problem even at larger scales:
>
> | Family | Size | L0 $V_p$ | L0 $V_s$ | Gap | L1 $M$ |
> |---|---|---|---|---|---|
> | Baseline | 0.4B | 93.5 | 39.8 | 53.6 | 47.8 |
> | Gemma-3 | 1B | 33.4 | 1.0 | 32.4 | 0.2 |
> | Gemma-3 | 4B | 100 | 35.3 | 64.7 | 0.0 |
> | Llama-3.2 | 1B | 96.9 | 38.4 | 58.5 | 0.8 |
> | Llama-3.2 | 3B | 97.8 | 49.0 | 48.8 | 1.0 |
> | Llama-2 | 7B | 98.4 | 33.9 | 64.5 | 33.1 |
> | Mistral | 7B | 96.0 | 49.9 | 46.1 | 23.3 |
> | OLMo-2 | 1B | 93.0 | 33.5 | 59.5 | 7.0 |
> | OLMo-2 | 7B | 93.5 | 28.0 | 65.4 | 14.3 |
> | Qwen-3.5 | 4B | 99.2 | 87.0 | 12.2 | 2.4 |
> | Qwen-3.5 | 9B | 99.7 | 89.8 | 9.9 | 0.5 |
>
> The $V_p$–$V_s$ gap persists across all models and scales (9.9–65.4pp). It varies more by model family and tokenizer design than by parameter count. OLMo-2 shows 60–65pp gaps at both 1B and 7B. Even the best model (Qwen-3.5 9B) retains a 9.9pp gap. These results show the phenomena we document are not specific to our baseline - they appear across widely used open-weight models up to at least 9B parameters.
>
> ## Weakness 2. Constrained generation solves this problem
>
> We acknowledge we should have cited constrained decoding (Koo et al., 2024; Willard & Louf, 2023, Cognetta and Okazaki 2025) and will do so in CR upon acceptance.
>
> However, constrained decoding addresses structural validity - ensuring generated bytes form valid UTF-8. Our paper studies a different question: whether models learn to produce the correct character, not just valid bytes. These are distinct failure modes, as our data shows.
>
> Analyzing cases where $\Delta_{LL} > 0$ (model assigns higher likelihood to the correct completion but greedy decoding fails), we find that in 100% of such cases the model already produces structurally valid continuation bytes — just the wrong ones. For example, after lead byte 0xE3 the model always emits 0x80 (producing CJK Symbols) instead of the target character. These outputs satisfy the DFA constraint. Constrained decoding would not change them. Term match rates would remain unaffected.
>
> Our evaluation framework measures both structural validity ($V_{\text{partial}}$, $V_{\text{strict}}$) and semantic correctness (term match). Constrained decoding closes the former gap but leaves the latter untouched. Understanding why and when this semantic gap arises during training is the contribution of this paper, and it is complementary to constrained decoding rather than obviated by it.
>
> We understand that byte-level validity occupies a small corner of the broader LLM landscape. But the structural-semantic dissociation we document –  models learning valid byte patterns
> without learning correct byte content — is a general phenomenon that appears across model families, scales, and tokenizers. We hope you can  reconsider the significance of this
> finding in light of the cross-model evidence presented above.

---

> > ### Author Rebuttal · Reviewer_BL2Q · 2026-04-06
> >
> > To what extent is semantic validity similar to or different from the issue of selecting the right next token for rare tokens in contexts that are not in the middle of a Unicode character? Does their rate of convergence differ greatly, or do they both behave similarly in converging more slowly than perplexity?

---

> > > ### Author Response · Authors · 2026-04-08
> > >
> > > Thank you for the response and follow-up. Our understanding is that the question is asking whether the slow convergence of byte-level semantic completion is a unique phenomenon or just a special case of the general difficulty models have with rare information.
> > >
> > > While we did not run a controlled head-to-head comparison, we acknowledge a partial overlap: the "2x lag" claim compares byte-level validity against overall perplexity, which is dominated by common tokens. To that extent, "validity lags perplexity" is partly "the long tail lags the common case," which is not unique to byte-aware models.
> > >
> > > With that said, two observations from our data are not consistent with generic rare-token convergence. First, the failure mode is concentrated mode collapse, not distributed uncertainty: in 100% of $\Delta_{LL} > 0$ failures, the model emits the same byte for each lead byte (0xE3 -> 0x80, 0xEC -> 0x97, 0xF0 -> 0x9F). Generic rare-token failures show a spread of plausible alternatives, not a single dominant mode. Additionally, in our cross-model evaluation, tokenizer setup overrides scale: small vocabularies (or large vocabularies trained with high linguistic bias) that force frequent byte-fallback (Our small model at 47.8%, Llama-2 7B 33.1%, Mistral 7B 23.3%) outperform much larger models with large vocabularies (Qwen-3.5 9B 0.5%, Gemma-3 4B 0.0%). If this were generic rare-token learning, we would likely not have observed this pattern.
> > >
> > > While these observations suggest distinct dynamics at the byte level, definitively establishing whether this convergence is quantitatively different from rare-subword learning requires a controlled comparison. That said, we do believe that the work identifies fundamanetal problems in a frequently overlooked space, which is related but not strictly coupled with infrequent characters, especially as it depends on a strict structural restriction to produce valid sequences. There is a bit more to investigate in this direction, and future work should track per-cohort cross-entropy on subword tokens versus byte-fallback sequences sampled at equal training frequencies. A further ablation training two otherwise-identical models differing in CJK vocabulary coverage should further demonstrate the tokenizer setup being a strong contributor to this effect.

---

### Decision · Program_Chairs · 2026-04-30

**Decision:**

Accept (regular)

**Comment:**

This paper received mixed reviews, but the balance of the discussion is cautiously positive. Reviewers agreed that the paper is technically sound, clearly written, and addresses a real reliability issue for byte-level language models that is not well captured by perplexity alone. In particular, the DFA-based partial-credit metric and the separation between structural validity and semantic correctness were viewed as useful contributions, and several reviewers found the training-dynamics study insightful.
The main reservations concern significance and empirical breadth. Two reviewers questioned whether UTF-8 validity is central enough for ICML, especially given the availability of constrained decoding, and one reviewer remained concerned that the strongest claims are based primarily on a single 355M model. There was also discussion about whether the observed lag is specific to UTF-8 validity or simply another instance of slower learning on rare-token phenomena. The rebuttal helped materially here by adding evidence on stronger contemporary models and by resolving the main concerns of the more positive reviewers. Although I do not view this as a consensus accept, I believe the paper identifies a concrete and underexplored failure mode and provides a useful evaluation framework that others can build on.